# Acceptance of Illness and Compliance with Therapeutic Recommendations in Patients with Hypertension

**DOI:** 10.3390/ijerph17186789

**Published:** 2020-09-17

**Authors:** Agnieszka Pluta, Beata Sulikowska, Jacek Manitius, Zuzanna Posieczek, Alicja Marzec, Donald E. Morisky

**Affiliations:** 1Department of Preventive Nursing, Faculty of Health Sciences, Ludwik Rydygier Collegium Medicum, Nicolaus Copernicus University, 87-100 Toruń, Poland; alkam7@o2.pl; 2Department of Nephrology, Hypertension and Internal Diseases, Faculty of Medicine, Ludwik Rydygier Collegium Medicum, Nicolaus Copernicus University, 87-100 Toruń, Poland; beata.sulikowska27@wp.pl (B.S.); nerka@nerka.cpro.pl (J.M.); 3Department of Endocrinology and Diabetology, Provincial Children’s Hospital Józef Brudziński, 85-667 Bydgoszcz, Poland; zuzannaposieczek@icloud.com; 4Department of Community Health Sciences, UCLA Fielding School of Public Health, Los Angeles, CA 90095, USA; dmorisky@ucla.edu

**Keywords:** arterial hypertension, adherence, compliance, acceptance of illness

## Abstract

Arterial hypertension (AH) is one of the most common cardiovascular diseases increasing mortality rates in Poland and worldwide. Due to its prevalence, complications and treatment costs, AH is a significant health-related, economic and social problem. The aim of this study was to assess the level of acceptance of illness and compliance with therapeutic recommendations in patients with AH. The study included 200 outpatient hypertensive patients, 85 men and 115 women aged 49.1 ± 11.6, and used the standardized acceptance of illness (AIS), the eight-item Morisky Medication Adherence Scale (MMAS-8) and author’s design questionnaires. The level of acceptance of illness was found to be as follows: higher in men than in women, unaffected by comorbidities or sociodemographic factors such as residence and professional activity, decreasing with age, and correlating negatively with the duration of antihypertensive therapy. The level of adherence and compliance did not affect the AIS score and increased with the level of education. The study population demonstrated an overall good level of acceptance of illness. Men were characterized by lower levels of adherence and compliance. Patients with AH presented a moderate level of adherence and compliance, which indicates the need for providing active education, support and extensive cooperation facilitating their conformity to therapeutic recommendations.

## 1. Introduction

Arterial hypertension (AH) is a disease of civilization. The number of adults with high blood pressure increased from 594 million in 1975 to 1.13 billion in 2015, mainly in low- and middle-income countries [1]. According to the National Health and Nutrition Examination Survey (NHANES), in the United States AH is found in 33.5% of the adult population [2]. In Poland, the results of the Multi-Center National Population Health Survey WOBASZ II of 2014 indicate that 14 million Poles aged 19–99 suffer from AH [3]. In a recently published study involving a sample of nearly 6000 individuals, AH was observed in 35.2% of the study population over 18 years of age [4]. Its incidence increases with age. AH is a risk factor for serious heart, kidney and brain diseases, which is why an early diagnosis and a properly implemented therapy are crucial for the future of the patient [5,6]. Unfortunately, the effectiveness of AH treatment in Poland is still insufficient. Generally, 46.1% of patients with hypertension receive treatment and only 23% of patients achieve adequate pressure control [3]. It is not enough to take medication regularly and it is necessary to follow a treatment regimen for a long time. Non-pharmacological, targeted therapy dealing with modifiable risk factors is an inseparable element of AH therapy. It requires commitment on the part of the patient, participation in the treatment process and a change of existing habits and routines [6]. In order to achieve positive treatment results, it is necessary to respect therapeutic recommendations and take medication systematically (compliance), as well as cooperate with medical staff (adherence) [7,8]. Non-compliance with therapeutic recommendations, which is a particularly common phenomenon among chronically ill people, especially older adults [9], also has economic consequences for both the patient and the health care system [10,11]. Failure to comply has a complex cause. Compliance with recommendations is not only dependent on the patient. It is a process in which the patient himself, their relatives and health care providers are involved. Factors conducive to compliance with therapeutic recommendations include: female sex, university education, wealth, having a family or a spouse, good relationship with one’s doctor, patient involvement, support through appointment reminders (telephones, e-mails and text messages), drug dispensers, blood pressure self-monitoring, simple therapy regimen, use of combined preparations, goals of antihypertensive therapy understandable to the patient and concomitant diseases, especially past cardiovascular complications or persistent high blood pressure [10,11,12,13,14].

Factors adversely affecting compliance with therapeutic recommendations attributable to medical staff include too little time spent with the patient, failure to provide understandable instruction on the use of medications, showing no interest in the degree of compliance with therapeutic recommendations by the patient and unprofessional conduct and failure to fulfill the professional role of a medical worker, especially the educational and supportive role [12,13].

When analyzing patients’ compliance with therapeutic recommendations, one cannot ignore the factors resulting from a given disease entity. The problem of non-compliance is more often related to patients with chronic, mild-symptom or asymptomatic diseases [12,13,14,15]. Often, patients who are diagnosed with a chronic disease adhere to recommendations initially, but over time their motivation decreases and they increasingly withdraw from following them. The impact of demographic characteristics on compliance with therapeutic recommendations is often underestimated. It is worth emphasizing that the age of the patient adversely affects the intellectual performance, which in turn reduces the ability to follow medical recommendations [16].

Another important factor determining a patient’s adherence to the therapeutic plan is the acceptance of their illness. It is a complex psychological phenomenon, which is of a constructive significance. It can be positive, neutral or negative. Studies on acceptance of an illness are qualitative in nature and are used to assess a patient’s adaptation to the disease and related limitations. Acceptance of an illness is always a subjective and very individual measure [15]. Positive perception of one’s own health status improves compliance and adherence of the patient. An assessment of the acceptance of the disease makes it possible to identify the needs and problems specific to the patient with AH, and thereby optimize and personalize the therapy, which can in turn contribute to a higher level of adherence.

The aim of the present study was to assess the level of acceptance of illness and compliance with therapeutic recommendations in patients with hypertension. The impact of sociodemographic and clinical factors on the degree of disease acceptance was also assessed.

## 2. Material and Methods

### 2.1. Subjects

The study group consisted of 200 individuals diagnosed with arterial hypertension, including 85 men and 115 women. The examination was carried out at the “GRYF-MED” Multidisciplinary Healthcare Center in Bydgoszcz and the Nephrology and Hypertension Clinic at the Antoni Jurasz University Hospital No. 1 in Bydgoszcz.

Criteria for inclusion in the study were as follows:Clinically diagnosed hypertension,Minimum age of 18 years,Consent to voluntary participation.

The study was conducted using an anonymous questionnaire of the authors’ own design, containing 16 questions and providing basic information about sociodemographic factors and the disease entity in question, hypertension, as well as 17 questions related to lifestyle, methods of AH treatment and health care received.

Standardized research tools have also been used:Questionnaire for assessing the acceptance of illness using the acceptance of illness scale (AIS) by Felton et al., Polish adaptation by Zygfryd Juczyński [17]. The questionnaire contains eight statements describing the negative consequences of poor health: problems adapting to restrictions imposed by the disease, limitations on performing favorite activities, a sense of inadequacy, a sense of dependence, a sense of being a burden to loved ones, reduced self-esteem, reduced self-sufficiency and a sense of awkwardness, among others. The answers are given on a 5-point Likert scale, where 1 stands for poor adaptation to the disease, while 5 stands for full acceptance of the disease. The total score is in the range of 8–40 points.

For the purposes of this study, the study population was divided into three groups, depending on the degree of acceptance of illness: group I—no AIS (8–18 points), group II—moderate AIS level (19–29 points) and group III—good AIS level (30–40 points).

The MMAS−8 diagnostic adherence assessment instrument was the questionnaire for assessing the level of cooperation and compliance with medical recommendations (adherence and compliance). Respondents were asked to address 8 statements regarding their medication taking routine. The total result obtained for all statements allows one to assess the degree of compliance with taking recommended drugs by respondents, where a score < 6 points indicates a low level of drug compliance, 6 < 8 points scored indicate a moderate level of drug compliance, and 8 points scored indicate a high level of adherence and compliance [18,19,20].

The paper-based questionnaires were handed in person and completed in the presence of the author.

### 2.2. Study Protocol

The study was carried out in accordance with the Declaration of Helsinki. The protocol of the study was approved by the Bioethics Committee of the Nicolaus Copernicus University Collegium Medicum in Bydgoszcz, Poland (KB 365/2017). The survey was anonymous.

### 2.3. Statistical Analysis

Statistical analysis was performed using the Statistica v. 10 software and the Excel spreadsheet (StatSoft, Poland, Kraków) (. Basic qualitative (nominal) data were presented as the population size (*n*) and percentage (%). The significance of correlations between qualitative variables was verified using the Pearson’s chi-square test at the level of *p* < 0.05. Measurable (quantitative) variables were presented as mean and standard deviation (M ± SD). For variables with non-normal distribution, the median and range were given. Analysis of variance was used to determine the significance of differences between more than two groups.

## 3. Results

The general characteristics of the study group are presented in Table 1. The study population consisted of 200 participants, including 85 men and 115 women. The mean age of the subjects was 49.1 ± 11.6 years. The youngest respondent was 18 years old and the oldest was 72 years old. Over a half of the surveyed population (137 respondents, 68.5%) were city residents. As many as 85 respondents (42.5%) had secondary education, 57 (28.5%) completed vocational training, 47 (23.5%) had higher and 11 (5.5%) had primary education. Over a half of the surveyed population (126 respondents, 63%) were economically active, 50 (25%) received retirement or disability benefits, 3 (1.5%) were students and 21 (10.5%) were unemployed. The mean duration of hypertension was 8.7 ± 7.4 years. Over 80% (162) of the respondents lived with their families. In the study population, 87 patients (43.5%) declared coexisting diseases. When answering the question about the type of comorbidity, some respondents marked several statements. A total of 87 people provided 117 responses. Most respondents (45 people) indicated the occurrence of diabetes, which constituted 38.5% of all the responses selected. The subjects further indicated hypothyroidism—15 people (12.8%), asthma—8 people (6.8%) and degenerative diseases—5 people (4.3%). One comorbid disease was indicated by 64 people (32%), two diseases by 20 people (10%), three diseases by 2 people (1%) and seven comorbidities by 1 person (0.5%).

The breakdown of age groups for sex is presented in the Figure 1.

The mean disease acceptance score was 30.55 ± 7.0 points. The study population was divided into three groups depending on the degree of acceptance of their illness. The largest group included individuals with a high level of acceptance of their illness and it included 124 people (62.0%). Ten people (6.0%) did not accept their illness (Table 2).

Table 3 presents the mean values for individual statements of the AIS scale, which range from 3.5 to 4.11 points. The lowest mean score, and thus the lowest level of acceptance of illness was reported for the statements: “I have trouble adapting to the restrictions imposed by the disease” (3.5 points) and “because of my condition I am not able to do what I like most” (3.68 points). The highest mean value, and thus the highest average degree of acceptance of their illness were reported for the statements: “I think people staying with me often feel awkward because of my illness” (4.11 points) and “the disease makes me a burden to my family and friends” (3.92 points).

A detailed set of the disease acceptance level results is presented in Table 4. Statistical analysis showed a higher level of disease acceptance in men than in women (31 ± 5.74 vs. 30.22 ± 7.83 points; *p* = 0.036). Nearly 30% of men and 35.7% of women showed a moderate level of acceptance of their illness. Those found not to accept their disease were 7.8% of women and 1.2% of men. Those that showed a good acceptance of their illness were 56.5% of the female and 69.4% of the male population.

Statistical analysis showed that the level of disease acceptance decreased with age (*p* = 0.01). The highest mean score for disease acceptance was reported in the age group of up to 30 years (34.41 points) and in the age group of 31–40 years (33.3 points). The lowest score was determined in the age group of 41–50 (29.63 points) and over 60 (27.67 points).

The level of disease acceptance was found not to be determined by the place of residence of the respondents (*p* = 0.102).

The results of the study demonstrated no correlation between the level of education of the respondents and their level of disease acceptance (*p* = 0.108). Of those that showed a high level of acceptance of illness, 70.2% of those had higher education, 62.4% had secondary education, 57.9% had vocational education and 45.5% had primary education. A lack of acceptance of their illness was demonstrated in 2.1% of respondents with higher education, 7.1% with secondary education, 3.5% with vocational education and 9.1% with primary education. A moderate level of disease acceptance was found in 27.6% of persons with higher education, 30.6% with secondary education, 38.6% with vocational education and 45.5% with primary education.

There were statistically significant differences between the groups of professional activity regarding the disease acceptance level (0.004). The highest mean disease acceptance score was recorded in the group of working patients—32.27 points.

Statistical analysis showed a significant negative correlation between the period of AH therapy and the level of disease acceptance (*p* = 0.020). The highest level of acceptance of the disease was found in individuals who suffered from AH for less than 3 years (32.88 points) and between 6 and 10 years (30.84 points), and the lowest in the group of patients whose disease lasted over 10 years (28.18 points). In this group, 67.5% of respondents declared a good level of AH acceptance, while the remaining ones (32.5%) reported a moderate level of acceptance of the disease. In the group of patients suffering from AH for 6–10 years, a good level of disease acceptance was found in 66.7%, moderate in 21.7% and low in 4.8% of the respondents. The lowest level of acceptance of the disease was demonstrated by those who were ill for over 10 years. In this group, 45.1% were people with a good level of disease acceptance, and 49% were patients with a moderate level of AH acceptance.

Statistical analysis showed no correlation between the occurrence of coexisting diseases in the respondents and their level of disease acceptance (*p* = 0.262). A higher level of disease acceptance was reported among patients without than in those with comorbidities, but it did not reach statistical significance (31.12 vs. 29.8 points; *p* = 0.26).

Adherence and compliance were assessed on the basis of the MMAS−8 questionnaire. Answers to individual questions of the MMAS−8 questionnaire are presented in Table 5. The mean number of points obtained was 6.48 ± 1.5, which indicated a moderate level of adherence and compliance.

The most numerous group of respondents were people with a moderate level of adherence and compliance—72 individuals (36%). Among the respondents, 66 people (33%) reported a low level and 62 people (31%) reported a high level of adherence and compliance.

Table 6 presents the detailed results of adherence and compliance. The highest mean score for the regularity of drug use was found in the age group of up to 30 years—6.74 points and over 60 years—6.65 points, while the lowest level was determined in the age group of 41–50 years—6.29 points. Along with an increase in the level of education, the level of adherence and compliance increased in patients (*p* = 0.001). A higher mean adherence and compliance score was reported in the female group—6.78 points, while the mean score in the male group was 6.08 points (*p* = 0.001).

There were statistically significant differences between the groups of professional activity regarding the level of systematic drug use (*p* = 0.006). The highest point mean of the systematic use of drugs was recorded in the group of retirees/disability pensioners—6.87 points and the lowest in the group of others (students, unemployed)—5.46 points.

The number of comorbidities did not correlate statistically with the level of adherence and compliance (*p* > 0.05). The highest mean point of systematic use of drugs was recorded in the group with two or more comorbidities—6.87 points and the lowest in the group without comorbidities—6.37 points.

The age of subjects, duration of AH therapy and place of residence were not statistically significantly correlated with adherence and compliance results (*p* > 0.05). There was no statistical significance between the level of systematic drug use and disease acceptance. Subjects with high adherence and compliance results presented slightly the highest level of disease acceptance. The lowest level was presented by respondents with poor results related to systematic drug use (Figure 2).

## 4. Discussion

Hypertension is one of the major cardiovascular risk factors. Adherence to therapeutic recommendations and cooperation between the patient and the therapist constitutes the basis for effective treatment of chronic diseases. Compliance with recommendations by patients with a chronic disease such as AH is unsatisfactory [13,15,21]. This problem is acknowledged all over the world by both physicians and the patients themselves [21].

Poor compliance and adherence are one of the main factors behind the failure of antihypertensive therapy in patients with AH. Acceptance of the disease is not without significance. A positive perception of one’s own health situation improves patient’s compliance and adherence. Acceptance of illness allows a patient to function properly despite the various risks, limitations and problems posed by a chronic disease. Awareness of its causes and effects, as well as knowledge of possible complications allows patients to gain effective self-control and implement healthy behaviors in order to prolong their life and improve its quality [16]. Acceptance of illness is a phenomenon studied from the perspective of social and clinical psychology and sociology, and it is sometimes underestimated by the medical community [14].

Teams of specialists dealing with chronically ill patients with AH do not always include a clinical psychologist. At the primary health care level, where the recipient of therapy is a patient with AH, consultation with a clinical psychologist is not readily available.

Acceptance of illness can have a positive effect on the patient’s adjustment to therapeutic recommendations as well as on the level of cooperation with medical personnel [8,9,10]. Regular use of medication, follow-up visits to the doctor, keeping a self-monitoring diary, implementation of new recommendations, as well as other factors such as comorbidities, addictions or age are all reflected in the patient’s acceptance of their current state of health [8,16,20,21,22].

Carrying out our own study and analyzing its results allowed us to assess the level of acceptance of a chronic disease as exemplified by patients with AH using the AIS scale and to examine the impact of factors affecting a given level of disease acceptance. The most numerous group, comprising of 62.0% of the respondents, were individuals with a high level of disease acceptance, which is consistent with the results of other researchers [23]. The available literature contains reports from studies where the highest percentage of patients with AH (from 57.8% to 86.8%) was found to be characterized by a moderate level of disease acceptance [24,25]. These discrepancies can be related to both the age of the patients and their state of health. In our study, the mean age of patients was 49.1 years, and the study itself was conducted among patients with AH treated in the outpatient setting, while other researchers conducted the study among patients with AH in the hospital setting [23,24]. It is known that the degree of disease acceptance is associated, among other things, with the functional state and mood of the patient as well as the severity of the disease [22,26].

Acceptance of an illness is a process that consists of many stages, such as shock, confrontation, escape and assimilation, and which depends on many factors. Everyone is different, so the reaction to the disease and acceptance of the new situation is also different [27].

The analysis of the results of our own research showed that sociodemographic factors such as sex, age and duration of antihypertensive therapy affected the degree of acceptance of their illness. Based on the data obtained, it was found that men showed a higher level of acceptance of their health than women (31.0 ± 5.7 vs. 30.2 ± 7.8 points; *p* = 0.036), which was reflected in the studies involving patients receiving outpatient treatment (*n* = 154; 27 ± 7.24 vs. 24 ± 8.25; *p* < 0.05) [25]. However, a different result was observed in a group of hospitalized patients, where a higher level of disease acceptance was found in the female population [28]. In another study involving 105 patients with AH, no statistically significant correlation was found between the level of disease acceptance and sex [23].

The decrease in the level of disease acceptance with age demonstrated in the statistical analysis was confirmed by other researchers [25]. The patient’s age reduces the ability to follow medical recommendations due to an adverse effect on their intellectual performance [16].

The analysis showed that the longer the duration of AH, the lower the level of disease acceptance (*p* = 0.02). The above relationship was confirmed by researchers who showed that patients treated for AH for 1–3 years showed better disease acceptance rates than those treated for 3–5 years (27.6 ± 8.5 vs. 23 ± 6.51 points; *p* < 0.05) [25].

The results of the MMAS−8 questionnaire used in this study show a tendency for the level of acceptance of their illness to increase along with compliance with therapeutic recommendations, though it did not reach a statistically significant level, which may result from the size of the study population.

In this study, no relationship was found between the level of acceptance of their illness and such factors as education, place of residence or the occurrence of comorbidities. The above cited team of researchers [23] observed a statistically significant correlation of ischemic heart disease (*p* = 0.003), atherosclerosis and vascular diseases (*p* < 0.001) with the level of disease acceptance. Moreover, the incidence of those diseases was significantly lower in patients with good AIS levels.

In the present analysis, no statistically significant correlation was found between acceptance of their illness and the level of adherence and compliance assessed on the basis of the MMAS−8 questionnaire. The mean number of points obtained was 6.48 ± 1.5, which indicated a moderate level of adherence and compliance.

The analysis of the results of our own research showed that sociodemographic factors such as sex and education influenced the level of adherence and compliance. Women showed a higher level of adherence and compliance than men (6.78 ± 1.39 vs. 6.08 ± 1.57 points; *p* = 0.001). Along with an increase in the level of education, the level of adherence and compliance increased as well (*p* = 0.001). The highest level of the regularity of drug use was reported in the group of subjects with a secondary education (6.86 points) and higher education (6.64 points).

In this study, age, duration of AH therapy and place of residence did not determine the level of adherence and compliance. A higher level of adherence in older patients with AH as compared to younger people was demonstrated in a recently published meta-analysis, where adherence in people over 60 was associated with age, socioeconomic status and therapy-related factors [29].

Non-compliance with medical recommendations contributes to poor blood pressure control and applies to a large number of patients. Persistently high pressure values lead to impaired vascular function, which is associated with an increased risk of cardiovascular disease [30,31,32]. Intensive SBP (systolic blood pressure) control (target, <120 mmHg) following a microsimulation model covering the effects of systolic blood pressure intervention trial (SPRINT) treatment in adults at high risk of cardiovascular disease but without diabetes during a 5-year period prevented cardiovascular events and prolonged life [19].

The role of cooperation with medical staff should be emphasized when working with a patient with AH. Many doctors focus only on the negative effects of non-compliance. One of the biggest problems faced by the healthcare system in Poland is the acute shortage of medical staff at both primary and specialist care levels. According to the data contained in the OECD 2019 report, there are 2.4 doctors per 1000 inhabitants and 5.1 nurses per 1000 inhabitants, which are the lowest rates among all EU countries [33]. Currently, cases are observed when a patient due to limited access to doctors, especially medical specialists, decides to modify their therapy, and sometimes even to resign from it entirely [12,13,16].

Adherence to therapeutic recommendations by patients with hypertension is the basis for the effectiveness of antihypertensive treatment. Information and communication technologies play an important role in improving adherence [21,31,34,35]. New methods of communication between the patient and the doctor are beneficial in terms of saving time, greater treatment effectiveness, avoiding medical errors and adverse situations, which was emphasized in the e-Health for Safety report published by the European Commission in 2007 [36]. Patients can use mobile phones, smartphones or many online portals. Easy access to various information technologies improves the patient’s contact with the doctor, blood pressure monitoring and, as a result, the effectiveness of treatment. It is especially important recently, during the COVID-19 epidemic. The epidemiological situation has contributed to a change in the organization of patients’ personal visits to primary healthcare and specialist clinics. When preparing a plan of work with a patient, one should take into account the individual characteristics and capabilities of the patient, including their attitude towards their health condition and acceptance of the disease. A psychologist would be helpful to the clinician in determining the acceptance of the disease. The supporting and educational role of medical personnel should be carefully performed.

Currently, a system of electronic medical records is being implemented in Poland and monitoring systems for the patient’s condition developed by individual clinical units are being inspected. There is no general system available to the public. The AH patient monitoring system has not been formally launched. For the efficient functioning of system monitoring the patient’s condition and his participation in the therapy, it is advisable to prepare both the patient and the medical staff.

The patient should be convinced to use online medical services (telemedicine) and be substantively and practically prepared in the area of use of electronic devices for health monitoring and virtual contact with a therapist. It may also be necessary to provide the patient with monitoring equipment with the appropriate technical parameters to ensure data collection and data transfer.

It is advisable to train medical and pharmaceutical personnel in the area of use of information technology and standards of communicating with the patient, which determines better patient’s understanding and compliance with therapeutic recommendations and, as a result, may contribute to improving blood pressure control and reducing cardiovascular risk.

## 5. Summary

Patients with a chronic disease such as hypertension are those who particularly require motivation to engage in the treatment process as well as professional support in adhering to therapeutic recommendations. Individual assessment of the degree of acceptance of their illness is of great importance for the clinician/therapist when preparing the patient for self-care and self-control.

Low levels of adherence and compliance among hypertensive patients increase the risk of cardiovascular complications.

At present, there is no standard developed that could solve the problem of non-compliance with therapeutic recommendations. The medical community is making great efforts to improve the current situation in this respect. These include, among other things, simplification of therapy regimens, education of the patient and their family, psychotherapy, crisis intervention and refresher education.

## 6. Conclusions

We demonstrated that patients with arterial hypertension presented a high level of acceptance of their illness determined using the AIS tool. The level of disease acceptance among hypertensive patients depended on sex and age. Men were characterized by a statistically significantly higher level of disease acceptance. The highest mean score for disease acceptance was reported in the age group of up to 30 years. Patients with arterial hypertension presented a moderate level of adherence and compliance. Adherence and compliance levels among hypertensive patients correlated with sex. Women were characterized by a statistically significantly higher level of regularity of drug use. The level of adherence and compliance among hypertensive patients also correlated with education. The highest mean score for the regularity of drug use was reported in the group of patients with secondary and higher education. The moderate levels of adherence and compliance suggest the need for active education, support and comprehensive cooperation with regard to following therapeutic recommendations by hypertensive patients.

## Figures and Tables

**Figure 1 ijerph-17-06789-f001:**
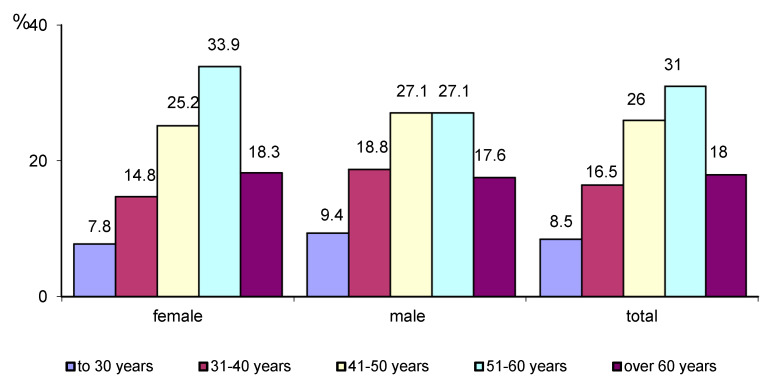
Age groups breakdown depending on sex.

**Figure 2 ijerph-17-06789-f002:**
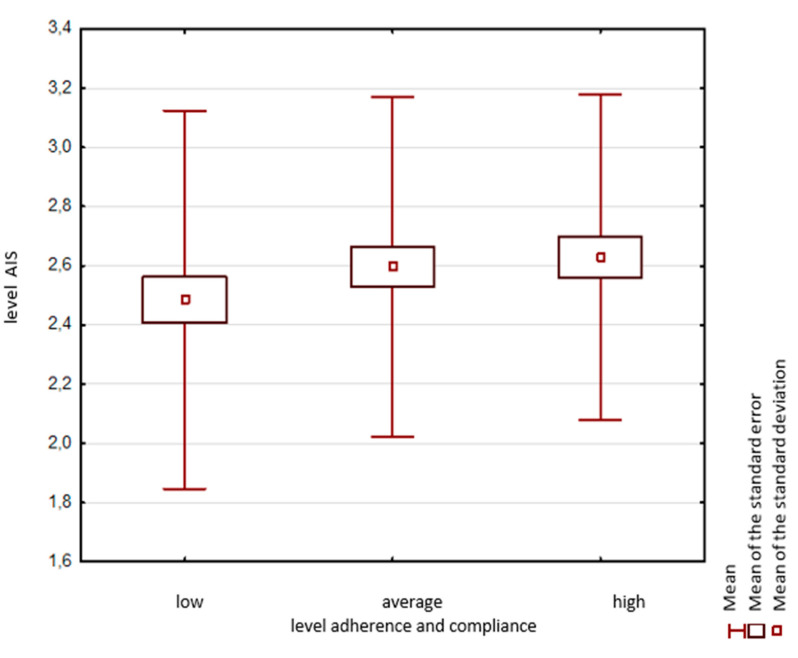
Distribution of AIS mean scores in adherence and compliance groups; Use of the ©MMAS is protected by US copyright and registered trademark laws. Permission for use is required. A license agreement is available from: MMAR, LLC., Donald E. Morisky, 294 Lindura Court, Las Vegas, NV 89138–4632; dmorisky@gmail.com.

**Table 1 ijerph-17-06789-t001:** The general characteristics of the study population.

Demographic Characteristics of Respondents	Group Size *n* = 200 (%)
Sex	
male	85 (42.5%)
female	115 (57.5%)
Age (in years)	
up to 30	17 (8.5%)
31–40	33 (16.5%)
41–50	52 (26%)
51–60	62 (31%)
over 60	36 (18%)
Place of residence	
country	63 (31.5%)
city	137 (68.5%)
Education	
primary	11 (5.5%)
vocational	57 (28.5%)
secondary	85 (42.5%)
higher	47 (23.5%)
Business activity	
unemployment	21 (10.5%)
full-time job	126 (63%)
disability pension/retirement	50 (25%)
pupil/student	3 (1.5%)
Living	
with family	162 (81%)
alone	38 (19%)
Coexisting diseases	
diabetes	45
hypothyroidism	15
asthma	8
degenerative diseases	5

**Table 2 ijerph-17-06789-t002:** Division of the study population into groups according to the level of acceptance of their illness.

Level of Acceptance	Points	Number	%
No acceptance	8–18 points	10	6
Moderate acceptance	19–29 points	66	33
High acceptance	30–40 points	124	62

**Table 3 ijerph-17-06789-t003:** Scores for acceptance of illness scale (AIS) statements.

Statement	Mean	SD	Confidence −95.0%	Confidence +95.0%	Median
1. I am having trouble adapting to the restrictions imposed by the disease	3.50	1.089	3.34	3.65	4.0
2. Because of my condition, I can’t do what I like most	3.68	1.155	3.52	3.84	4.0
3. The disease sometimes makes me feel unnecessary	3.76	1.249	3.59	3.93	4.0
4. Health problems make me more dependent on others than I would like to be	3.83	1.108	3.68	3.98	4.0
5. The disease makes me a burden to my family and friends	3.92	1.235	3.74	4.09	4.0
6. My condition makes me feel incomplete as a person	3.88	1.207	3.71	4.04	4.0
7. I will never be self-sufficient in the way I would like to be	3.89	1.104	3.73	4.04	4.0
8. I think the people around me often feel awkward about my illness	4.11	1.093	3.96	4.26	4.0

**Table 4 ijerph-17-06789-t004:** A detailed set of the disease acceptance level.

Variable		Level of Acceptance						*p*
		No Acceptance		Moderate Acceptance		High Acceptance		
		Number	%	Number	%	Number	%	
Age (in years)	up to 30	0	0.0	3	17.6	14	82.4	0.01
	31–40	0	0.0	7	21.2	26	78.8	
	41–50	3	5.8	22	42.3	27	51.9	
	51–60	3	4.8	20	32.3	39	62.9	
	over 60	4	11.1	14	38.9	18	50.0	
Sex	female	9	7.8	41	35.7	65	56.5	0.036
	male	1	1.2	25	29.4	59	69.4	
Education	primary	1	9.1	5	45.5	5	45.5	0.108
	vocational	2	3.5	22	38.6	33	57.9	
	secondary	6	7.1	26	30.6	53	62.4	
	higher	1	2.1	13	27.7	33	70.2	
Professional activity	disability pension/retirement	4	8.0	23	46.0	23	46.0	0.004
	full-time job	4	3.2	33	26.2	89	70.6	
	pupil/student/unemployed	2	8.3	10	41.7	12	50.0	
Place of residence	country	6	9.5	29	46.0	28	44.4	0.102
	city	4	2.9	37	27.0	96	70.1	
Living	alone	1	2.6	18	47.4	19	50.0	0.215
	with family	9	5.6	48	29.6	105	64.8	
Duration of the AH (in years)	up to 3	0	0.0	13	32.5	27	67.5	0.020
	3–5 l	4	8.7	10	21.7	32	69.6	
	6–10	3	4.8	18	28.6	42	66.7	
	over 10	3	5.9	25	49,0	23	45.1	
Coexisting diseases	yes	4	4.6	34	39.1	49	56.3	0.262
	no	6	5.3	32	28.3	75	66.4	
Number of coexisting diseases	none	6	5.3	32	28.3	75	66.4	0.326
	one	2	3.1	22	34.4	40	62.5	
	two and more	2	8.7	12	52.2	9	39.1	

**Table 5 ijerph-17-06789-t005:** Answers to individual questions of the MMAS−8 questionnaire *.

Question	Mean	SD	Confidence −95.0%	Confidence +95.0%	Median
1. Do you sometimes forget to take your medicine?	0.70	0.462	0.63	0.76	1.00
2. Sometimes people miss a dose for reasons other than forgetfulness. Have there been days in the last 2 weeks that you haven’t taken your medicine?	0.85	0.358	0.80	0.90	1.00
3. Have you ever reduced your dose or stopped taking the medicine without telling your doctor because you felt worse when taking it?	0.81	0.397	0.75	0.86	1.00
4. When you take a trip or leave the house, do you ever forget to take your medicine with you?	0.78	0.419	0.72	0.83	1.00
5. Did you take all your medications yesterday?	0.82	0.381	0.77	0.88	1.00
6. If you feel that your symptoms are under control, do you stop taking the medicine?	0.83	0.381	0.77	0.88	1.00
7. Daily use of medication is an inconvenience for some people. Have you ever felt that it is troublesome for you to follow your medication plan?	0.86	0.353	0.81	0.90	1.00
8. How often do you have difficulty remembering to take all your medicines?	0.55	0.281	0.51	0.59	0.50

* Use of the ©MMAS is protected by US copyright and registered trademark laws. Permission for use is required. A license agreement is available from: MMAR, LLC., Donald E. Morisky, 294 Lindura Court, Las Vegas, NV 89138–4632; dmorisky@gmail.com.

**Table 6 ijerph-17-06789-t006:** Detailed set of adherence and compliance results *.

Variable		Level of Adherence and Compliance						*p*
		Low		Average		High		
		Number	%	Number	%	Number	%	
Age (in years)	up to 30	4	23.5	8	47.1	5	29.4	0.613
	31–40	12	36.4	11	33.3	10	30.3	
	41–50	19	36.5	19	36.5	14	26.9	
	51–60	20	32.3	22	35.5	20	32.3	
	over 60	11	30.6	12	33.3	13	36.1	
Sex	female	27	23.5	44	38.3	44	38.3	0.001
	male	39	45.9	28	32.9	18	21.2	
Education	primary	9	81.8	2	18.2	0	0	0.001
	vocational	27	47.4	16	28.1	14	24.6	
	secondary	16	18.8	39	45.9	30	35.3	
	higher	14	29.8	15	31.9	18	38.3	
Professional activity	disability pension/retirement	11	22	19	38	20	40	0.006
	full-time job	41	32.5	46	36.5	39	31	
	pupil/student/unemployed	14	58.3	7	29.2	3	12.5	
Living	alone	14	36.8	11	28.9	13	34.2	0.91
	with family	52	32.1	61	37.7	49	30.2	
The time of the AH (in years)	up to 3	13	32.5	15	37.5	12	30	0.823
	3–5	16	34.8	14	30.4	16	34.8	
	6–10	21	33.3	26	41.3	16	25.4	
	over 10	16	31.4	17	33.3	18	35.3	
Number of coexisting diseases	none	40	35.4	39	34.5	34	30.1	0.529
	one	22	34.4	23	35.9	19	29.7	
	two and more	4	17.4	10	43.5	9	39.1	
AIS	low	5	7.6	3	4.2	2	3.2	0.394
	moderate	24	36.4	23	31.9	19	30.6	
	high	37	56.1	46	63.9	41	66.1	

* Use of the ©MMAS is protected by US copyright and registered trademark laws. Permission for use is required. A license agreement is available from: MMAR, LLC., Donald E. Morisky, 294 Lindura Court, Las Vegas, NV 89138–4632; dmorisky@gmail.com.

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
