# Peer review of "Acceptance of Illness and Compliance with Therapeutic Recommendations in Patients with Hypertension"

_ijerph, 2020, doi:10.3390/ijerph17186789_

Round 1
Reviewer 1 Report
In this manuscript titled, " Acceptance of Illness and Compliance with Therapeutic Recommendations in Patients with Hypertension ", Agnieszka Pluta et al., authors aimed to assess 19 the level of acceptance of illness and compliance with therapeutic recommendations in patients with arterial hypertension (AH). Overall, the manuscript is written clearly. However, the manuscript appears preliminary.
- Authors should clarify the patient’s gender. In abstract, they said 85 men and 115 women, but in results and table 1, there are 85 women and 115 men.
- Authors should improve the quality of figure 1, delete the “underline”.
- More volunteer participants should be recruited.
Author Response
We would like to thank all the reviewers (persons reviewing the work) for their thorough analysis of the work and substantive comments. Thanks to the attached comments, suggestions and tips, we were able to introduce changes to the work, and thus improve its substantive side. We will apply the remarks on the research methodology in future research in the group of people with hypertension.
Review
(2)
- Gender of patients was specified (85 men and 115 women).
- The quality of the table was improved and the "underlines" were removed.
- It was a pilot study. In the future, we plan to study the population of all patients with hypertension treated and consulted in cardiology and arterial hypertension clinics in the Kuyavian-Pomeranian Voivodeship.

Reviewer 2 Report
Authors present quite interesting study about arterial hypertension acceptance and compliance. 200 outpatient patients were asked through a questionnaire about their attitude to the disease and factors affecting it. Authors show quite good acceptance of the hypertension as well as level of compliance to the therapy. However:
1) please consider shortening the introduction, many data are not necessary or can be moved to the discussion section, like information about lack of medical staff workers (lines 69-74) or explaining the meaning of the acceptance of the illness (from line 87). Please try to focus on the main manuscript thesis in a short way.
2) please end the sentence starting from the line 92.
3) do you have information how many concomitant disorders patients had? please try to analyze compliance according to the number of comorbidities.
4) do you have information how many different drugs and how many tablets of drugs per day patients were taking?
5) if possible please try to extend the problem of electronic systems (lack of them?) and telemonitoring of patients with arterial hypertension, which is nowadays really important (lines 314-315).
6) also medical staff training in patients monitoring should be emphasized.
Author Response
We would like to thank all the reviewers (persons reviewing the work) for their thorough analysis of the work and substantive comments. Thanks to the attached comments, suggestions and tips, we were able to introduce changes to the work, and thus improve its substantive side. We will apply the remarks on the research methodology in future research in the group of people with hypertension.
Review
- The introduction was shortened and a lot of data was moved to the discussion section.
- The sentence starting at line 92 is completed.
- Information on comorbidities is provided in the results section.
- In the questionnaire, the authors included questions about drugs and the number of tablets taken daily. Patients, however, most often omitted these questions without answering, or indicated the answer "I don't remember". Due to the lack of complete data on medications and the number of tablets taken daily by patients, they were not included in further analysis.
- Information on electronic systems and telemonitoring of patients with AH was expanded in the discussion section.
- The discussion section covers the training of medical personnel in the area of patient monitoring.
Reviewer 3 Report
The paper describes the acceptance of hypertension and the adherence to therapies on a polish population of 200 subjects. Acceptance is higher in men, in youngers and in short therapy times. Adherence is lower in midle aged and higher in females.
Major points:
-Show results on tables (lines 180-211 and 222-235).
-Show data on acceptance according to business activity and living status.
-Show data on adherence according to education, bussines activity or living status.
Minor point:
Show age distribution according to sex.
Author Response
We would like to thank all the reviewers (persons reviewing the work) for their thorough analysis of the work and substantive comments. Thanks to the attached comments, suggestions and tips, we were able to introduce changes to the work, and thus improve its substantive side. We will apply the remarks on the research methodology in future research in the group of people with hypertension.
(3)
- The data described in lines 180-211 and 222-235 are presented in Tables 4 and 6.
- Data on the acceptance of the disease in terms of professional activity and life situation are presented.
- Data on adherence are presented in terms of professional activity, education, life situation.
4.The breakdown of age groups for sex is presented in the results section.
Round 2
Reviewer 2 Report
Authors sufficiently answered to all my questions and corrected the manuscript accordingly.